# Spatial and Temporal Dynamics of Chikungunya Incidence in Brazil and the Impact of Social Vulnerability: A Population-Based and Ecological Study

**DOI:** 10.3390/diseases12070135

**Published:** 2024-06-27

**Authors:** Thiago de Jesus Santos, Karina Conceição Gomes Machado de Araújo, Marco Aurélio de Oliveira Góes, Marcio Bezerra-Santos, Caíque Jordan Nunes Ribeiro, Allan Dantas dos Santos, Emerson Lucas Silva Camargo, Regina Claudia Silva Souza, Isabel Amélia Costa Mendes, Alvaro Francisco Lopes de Sousa, Damião da Conceição Araújo

**Affiliations:** 1Graduate Program in Nursing, Federal University of Sergipe, São Cristóvão 49100-000, Brazil; thiago.jesus.santos@hotmail.com (T.d.J.S.); maogoes@gmail.com (M.A.d.O.G.); caiquejordan_enf@yahoo.com.br (C.J.N.R.); allanufs@hotmail.com (A.D.d.S.); damiao.araujo92@gmail.com (D.d.C.A.); 2Medical and Nursing Science Center, Federal University of Alagoas, Campus Arapiraca, Arapiraca 57309-005, Brazil; kkkaraujo2006@yahoo.com.br (K.C.G.M.d.A.); marciobezerra.ufs@outlook.com (M.B.-S.); 3Ribeirao Preto College of Nursing, Universidade de São Paulo, Ribeirao Preto 14040-902, Brazil; lucmrg0@gmail.com (E.L.S.C.); iamendes@usp.br (I.A.C.M.); 4Hospital Sírio-Libanês, Faculdade Sírio-Libanês, São Paulo 01308-050, Brazil; regina.souza@hsl.org.br; 5NOVA National School of Public Health, Comprehensive Health Research Center (CHRC), NOVA University of Lisbon, 1600-560 Lisbon, Portugal

**Keywords:** chikungunya fever, social determinants of health, socioeconomic factors, spatial analysis, public health surveillance

## Abstract

To assess the temporal and spatial dynamics of chikungunya incidence and its association with social vulnerability indicators in Brazil, an ecological and population-based study was conducted herein, with confirmed cases of chikungunya and based on clinical and clinical–epidemiological criteria from 2017 to 2023. Data were obtained from the Notifiable Diseases Information System and social vulnerability indicators were extracted from the official platform of the United Nations Development Program and the Social Vulnerability Atlas. Temporal, spatial, and global spatial regression models were employed. The temporal trend showed that in 2017, the incidence increased by 1.9%, and this trend decreased from 2020 to 2021 (−0.93%). The spatial distribution showed heterogeneity and positive spatial autocorrelation (I: 0.71; *p* < 0.001) in chikungunya cases in Brazil. Also, the high-risk areas for the disease were concentrated in the northeast and north regions. The social vulnerability indicators associated with the outcome were those related to income, education, and housing conditions. Our analyses demonstrate that chikungunya continues to be a serious health concern in Brazil, but specially in the northeast and north regions. Lastly, mapping risk areas can provide evidence for the development of public health strategies and disease control in endemic regions.

## 1. Introduction 

Chikungunya fever is an acute arthropod-borne virus transmitted by *Aedes aegypti* and *Aedes albopictus* [1]. The virus is endemic in tropical and subtropical regions, but has a history of outbreaks and sporadic cases worldwide. Since its emergence in Tanzania in 1952, the virus has spread to areas of Africa, Asia, Oceania, and the Americas [2,3]. 

Chikungunya’s epidemiology is characterized by sporadic and unpredictable outbreaks in various countries. Globally, it is estimated that about 1.3 billion people living in 94 countries are at risk of infection by the virus. Importantly, India and Asia are among the most affected areas, facing significant public health challenges due to the high incidence of the disease. This worrying scenario can be related to various factors such as population density, lack of adequate infrastructure for mosquito control, poverty, and climatic conditions conducive to vector reproduction [4,5].

In Brazil, since the identification of the first autochthonous cases in 2014, chikungunya has emerged as a serious public health concern. The country has faced significant outbreaks exacerbated by favorable climatic conditions for vector proliferation, low social development index (SDI), and socio-environmental and economic vulnerabilities, especially in the northeastern region [1], with the state of Ceará being the most affected [6]. Several studies identified that indicators related to low education, inadequate water supply and sanitation, and low income and poverty were related to the incidence of arboviruses [7,8,9].

Regardless of acute symptoms, chikungunya also presents a concerning clinical aspect in its chronic phase. Many patients may develop late manifestations such as articulation pain that can last for months or years and limited body movement that significantly affects quality of life and social and emotional well-being [10,11].

Surveillance and vector control are essential to combat the resurgence and epidemics of arboviruses, including chikungunya [12]. In a continental country such as Brazil, this requires a multidimensional approach involving local governments, health agencies, research institutions, and the community to implement environmental, chemical, and biological control strategies and community interventions [13]. Considering this, Geographic Information System (GIS) and spatial analysis are useful tools for monitoring diseases like chikungunya. Also, GIS can be used to map the disease pattern and identify high-risk areas and, thereby, predict outbreaks and epidemics, as well as facilitate decision-making in arbovirus disease research and public health policies [14].

Despite the evidence of chikungunya’s impact, there are significant gaps in the knowledge of the spatial-temporal dynamics of the disease in Brazil. The geographical and temporal variation in the incidence of chikungunya, influenced by environmental, socioeconomic, and public health factors, still requires detailed investigation. This study may help researchers comprehend these gaps by providing critical insights into disease spread patterns and identifying high-risk areas. Furthermore, an understanding of these patterns is required for the development of more effective public health strategies for prevention and disease control. Thus, the objective of this study was to analyze the temporal and spatial dynamics of the incidence of chikungunya in Brazil and its association with social vulnerability indicators during 2017 and 2023.

## 2. Materials and Methods

### 2.1. Study Design and Area

An ecological, population-based, time-series study, in accordance with the guidelines provided by the Strengthening the Reporting of Observational Studies in Epidemiology (STROBE) [15] initiative, was conducted. Also, the study encompassed the 27 federal units of Brazil, including 26 states and the Federal District, as well as a total of 5.570 municipalities. To facilitate political and operational management, these states were categorized into five distinct regions: north, northeast, southeast, south, and central-west, each exhibiting its own geographical, economic, and cultural peculiarities (Figure 1).

### 2.2. Population, Case Definition, and Eligibility Criteria

The study population consisted of all notified and confirmed cases of chikungunya, as determined by laboratory and clinical–epidemiological criteria identified in each Brazilian municipality. The categorization of cases adhered to protocols established by the Brazilian Ministry of Health. A suspected case of chikungunya is defined as an individual presenting with abrupt fever higher than 38.5 °C, accompanied by intense arthralgia or arthritis not justified by other medical conditions, residing in or having visited endemic regions in the two weeks preceding the onset of symptoms, or who has an epidemiological link with a confirmed case [16].

A case confirmed by clinical–epidemiological criteria is one that meets the definition of a suspected case and has a familial or spatiotemporal link (epidemiological link) with a laboratory-confirmed case. A case confirmed by laboratory criteria is one that has obtained a positive laboratory result through viral isolation or viral RNA detection by RT-PCR (in a sample collected up to the 8th day from symptom onset), or detection of IgM antibodies in a single serum sample during the acute phase (from the 6th day of symptom onset) or convalescent phase (15 days after symptom onset), demonstration of seroconversion between acute phase samples (first sample) and convalescent phase samples (second sample), or detection of IgG antibodies in samples collected from patients in the chronic phase of the disease, with suggestive clinical presentation [16].

The criteria for inclusion in the study were as follows: (i) cases notified from 2017 to 2023; (ii) individuals residing in the municipalities of the states of Brazil; and (iii) data completed in the information system with a defined clinical classification. Records presenting with a clinical classification as unknown, blank, or inconclusive, as well as duplicate cases identified in the information system, were excluded from the analysis.

### 2.3. Variables and Data Sources

The outcome of the study was the incidence of chikungunya in Brazil. The covariates of interest were time and indicators of social vulnerability grouped into an index of human development, health, education, income, housing, and environment. The data on notified and confirmed cases of chikungunya were collected from the Notifiable Diseases Information System (SINAN) https://datasus.saude.gov.br/ (accessed on 14 October 2023). Data related to social vulnerability indicators were obtained from the United Nations Development Program (UNDP) http://www.atlasbrasil.org.br/ (accessed on 14 October 2023) and the Social Vulnerability Atlas (IVS) http://ivs.ipea.gov.br/ (accessed on 14 October 2023). The cartographic base of Brazil, available in the electronic database of the Brazilian Institute of Geography and Statistics (IBGE), was used for analysis, along with the corresponding cartographic projection of the Universal Reference System SIRGAS 2000.

### 2.4. Data Processing and Analysis

#### 2.4.1. Descriptive Analysis

The characterization of the population affected by chikungunya was performed ac-cording to sociodemographic variables. These variables were analyzed based on the ab-solute number of cases, relative percentage, and incidence rate. Incidence rates of chikungunya were calculated by regions of Brazil, gender, race/skin color, age group, and diagnostic criteria.

#### 2.4.2. Temporal Trend Analysis

A time series was constructed using the Prais–Winsten regression model. This model was utilized to account for the influence of first-order autocorrelation in the analysis of time series data. The data modeling process included transforming the rates (dependent variable = Y value) into a base 10 logarithmic function, and the outcomes of the logarithmic rates (β) from the Prais–Winsten regression allowed for the estimation of the annual percentage change (APC) for the region under study, along with their respective 95% confidence intervals. We calculated the probability values (*p*-value) and APC, considering a 95% significance level (95% CI) [10].

The incidence trends were classified as increasing (*p*-value < 0.05 and positive β), decreasing (*p*-value < 0.05 and negative β), or stable (*p*-value value ≥ 0.05). Stable trend means that there are no significant changes in the data over time [10].

#### 2.4.3. Spatial Analysis

Initially, the incidence rate calculation was performed by considering the total annual cases of chikungunya per municipality divided by the population of the municipality and multiplied by a constant of 100,000 inhabitants. To display the spatial distribution of chikungunya incidence, maps were constructed to visualize the incidence of chikungunya in Brazilian municipalities, along with maps showing the smoothed distribution of incidence rates using the local empirical Bayesian estimator. 

The local empirical Bayesian estimator was employed to minimize instability caused by the random fluctuation of cases by smoothing standardized rates through the application of weighted averages and creating a third corrected rate. The empirical Bayesian rate demonstrated a correction of the multiplicative rate equal to 1000, taking into account the population at risk and the number of cases for each analyzed year by municipal area [17].

To estimate spatial variability in the data analysis, a proximity matrix was constructed, in which neighboring and bordering municipalities were assigned a value of 1 (one), and those without adjacent bordering geometries were assigned a value of 0 (zero) [17].

Thematic maps were developed based on the calculation of chikungunya incidence in Brazilian federative units for the analyzed period. Spatial autocorrelation between rates was used to investigate whether the spatial distributions of the disease occur randomly or follow some pattern of occurrence in space. Thus, a spatial proximity matrix was created based on the contiguity criterion, adopting a significance level of 5% and calculating the Global Moran’s I Index ranging between −1 and +1, representing the expression of spatial autocorrelation of the disease in geographic space, analyzed to identify spatial clusters and define risk areas. Values close to zero indicate spatial randomness; values between 0 and +1 indicate positive spatial autocorrelation, and between −1 and 0, negative spatial autocorrelation [17].

The Moran scatterplot was used to indicate critical or transition areas, aiming to compare the value of each municipality with its neighbors and verify spatial dependence shown by the Local Index of Spatial Association (LISA) for detecting regions with significant spatial correlation [17].

This technique allowed for the graphical visualization of the degree of similarity between neighbors through the Moran scatterplot or Moran Map. This diagram consists of a two-dimensional plot of normalized values, z, by the average of the neighbors, wz. The diagram is divided into four quadrants: quadrants Q1 (positive values, positive averages) and Q2 (negative values, negative averages) indicate points of positive spatial association, in the sense that a location has neighbors with similar values. Conversely, quadrants Q3 (positive values, negative averages) and Q4 (negative values, positive averages) indicate points of negative spatial association, showing that a location has neighbors with different values [18].

#### 2.4.4. Global Spatial Regression Analysis

For this purpose, an approach integrating the spatial dimension was adopted to develop an explanatory model for the incidence of chikungunya, correlating it with social vulnerability indicators. The methodology used to identify the determinants of disease incidence involved the use of both non-spatial and spatial multiple linear regression models, following the steps described below [17].

First step: Initially, 100 social vulnerability indicators were collected. After conducting a detailed analysis of the database, we selected 26 variables that may be associated with the incidence of chikungunya. A descriptive analysis was performed, including the assessment of normality distribution using the Shapiro–Wilk test [17].

Second step: The Bayesian incidence rate of Chikungunya was transformed into a logarithmic function (Ln) to approximate the data distribution to a normal pattern. The 26 selected variables were correlated with the outcome using Spearman’s correlation. Variables that showed a correlation above 30% were included, totaling 14 variables (Appendix A). Subsequently, a collinearity analysis was conducted with the 14 variables, considering the variance inflation factor (VIF), selecting those with a VIF below 20. Other variables with a VIF between 10 and 20 were critically discussed among the researchers for inclusion in the model. As a result, 9 variables were retained [17].

Third step: In global spatial modeling, the ordinary least squares (OLS) method was adopted, with variable selection by the backward method. The determination coefficients and 95% confidence intervals were calculated. Moran’s spatial analysis of the residuals was used to determine the need to add a spatial component to the model. Upon identifying spatial dependence, Lagrange multiplier tests assisted in choosing between the Spatial Error Model and the Spatial Lag Model, following Luc Anselin’s guidance. In the Spatial Error Model, spatial effects are considered noise, while the Spatial Lag Model recognizes spatial autocorrelation in the response variable Y. The normality of residuals was tested by the Jarque–Bera test, and homoscedasticity by the Breusch–Pagan test [17].

Fourth step: The criteria for evaluating the model’s quality included the Akaike Information Criterion (AIC), the Schwarz Bayesian Criterion (BIC), the coefficient of determination (R^2^), the log likelihood, and Moran’s statistic for the residuals. The optimal model presented the lowest values of AIC and BIC, higher values of log likelihood and R^2^, and residuals with spatial independence [17].

#### 2.4.5. Resources and Software

The following programs were used for data processing and analysis: RStudio version 4.2.1 (www.rstudio.org), JASP 0.18.1.0 (jasp-stats.org), JAMOVI version 2.4.14 (www.jamovi.org/), GeoDa version 1.22 (Spatial Analysis Laboratory, University of Illinois, Urbana Champaign, Estados Unidos), QGIS version 3.4.11 (QGIS Development Team; Open Source Project of the Geospatial Foundation, CC BY-SA, Las Palmas, CA, USA).

## 3. Results

A total of 487.775 cases of chikungunya were confirmed in Brazil between 2017 and 2023, with an incidence of 232.1 per 100.000 inhabitants (Appendix A). We first assessed temporal trends and our data indicate an expressive increasing trend in 2017 with an APC (Annual Percentage Change) of 190% (95% CI: 0.96–2.48; *p* < 0.001). In 2018–2019, the trend was stationary, and in 2020–2021, it was decreasing, (APC = −0.93%; 95% CI: −1.18–−0.98; *p* < 0.0001). For 2022–2023, the trend was stationary (Table 1).

Table 2 presents the spatial autocorrelation of chikungunya incidence in Brazil from 2017 to 2023. All Global Moran’s Indices were significant (*p*-value < 0.05), ranging from 0.57 to 0.78, which indicates positive autocorrelation for each year and for the combined periods. Unusually, in 2020, the index is negative (−0.63; *p*-value < 0.05), suggesting an inverse spatial autocorrelation.

Additionally, the spatial analysis of chikungunya incidence revealed a high concentration of the disease in all areas of Brazil, mainly in the northeast and north regions. The concentration of municipalities with high endemicity in the states of Minas Gerais and Rio de Janeiro, both in the southeast region, also deserves attention. The application of the Bayesian estimator to smooth the rate resulted in reduced variation and in the identification of risk clusters in those regions (Figure 2A,B).

Moreover, the Moran’s Index calculated for the adjusted average rate was significantly positive during the studied period (I: 0.52; *p*-value < 0.01), indicating a spatial autocorrelation of chikungunya incidence. High risk clusters were observed in the northeast and north regions (Q1: high-high; *p* = 0.01). On the other hand, clusters with the lowest incidences were detected in the southeast, couth, and central-west (Q2: low-low; *p* = 0.001) (Figure 2C).

For the global spatial regression modeling, 14 out of 26 social vulnerability indicators showed a positive correlation with the outcome (Table 3). The correlation analysis showed that indicators with better socioeconomic conditions demonstrated an inverse correlation with the incidence of chikungunya in Brazil. Nevertheless, indicators related to the literacy rate and the percentage of people in households with inadequate water supply and sanitation showed a positive correlation with the outcomes. In addition, the indicator related to the vulnerable population aged 15 to 24 years (Rho = 0.45; *p*-value ≤ 0.01) and the population in vulnerable households with elderly people (Rho = 0.45; *p*-value ≤ 0.01) showed significant correlation with the incidence of chikungunya.

A collinearity test was applied for the evaluation of VIF < 20 and selection of variables for inclusion in the model. Nine (9) variables were selected (Table 4).

The analysis of the Moran’s Index residuals from the OLS model was significant for the disease: the index of the residual = 0.38 with a *p*-value = 0.01, indicating spatial dependence. The inclusion of the spatial component in the regression model was based on the Lagrange multiplier diagnostics for spatial dependence, which indicated the spatial lag and spatial error models. Herein, the spatial model was the best at explaining the factors associated with the incidence of chikungunya in Brazil, as observed by the lower values of AIC and BIC, as well as the higher values of R^2^ and log likelihood, confirming that the spatial lag model was more suitable for the analysis in question.

Applying the spatial lag model, an association was found between indicators of inequality and social vulnerability related to income, housing, and urbanization with the incidence of the disease. Herewith, those that showed the highest association were the social vulnerability index (R^2^: −1.28; *p* < 0.01), percentage of people in households with inadequate water supply and sanitation (R^2^: 0.80; *p* < 0.01), percentage of people aged 15 to 24 who do not study, do not work, and have a per capita household income equal to or less than half the minimum wage (R^2^: 0.16; *p* < 0.03), social vulnerability index—income and work dimension (R^2^: 1.93; *p* < 0.01), per capita income (R^2^: 0.94; *p* < 0.01), and the percentage of the population in households with a density >2 (R^2^: 0.58; *p* < 0.04) (Table 5).

## 4. Discussion

Chikungunya epidemics pose a significant challenge to global public health, but especially in low- and middle-income countries in tropical and subtropical areas. This disease substantially impacted health services and the life quality of patients [12,19]. This study was conducted in Brazil, a developing nation that has faced punctual epidemics of this disease and remains vulnerable due to its unique social, political, and environmental characteristics. Also, temporal and spatial analysis, and the correlation with social vulnerability indicators, were employed to comprehensively understand the epidemiological landscape of chikungunya since its onset in Brazil [19].

Our findings demonstrated an increasing temporal trend (APC = 1.90%) in the incidence of chikungunya in 2017. Importantly, during this period, Brazil experienced a notable rise in the number of disease cases compared to preceding years. According to information released by the Ministry of Health, approximately 185,000 suspected cases of chikungunya were reported nationwide in 2017. These data represent an meaningful increase and demonstrate the impact of the disease in Brazil. The northeast and southeast regions emerged as the most impacted by the disease [20,21].

We observed decreasing trends in the disease’s prevalence in Brazil during the 2020–2021 period. In contrast to these findings, a study employing Seasonal Autoregressive Integrated Moving Average (SARIMA) models projected 106,162 serologically confirmed cases for the 2020–2021 period and 339,907 reported cases for the 2022–2023 period, with a higher concentration of cases in the northeast and southeast regions of the country [22]. The decreasing trend identified in this study may be related to the emergence of the COVID-19 pandemic. Brazil confirmed its first case of the disease in February 2020, which started an extremely challenging period for national public health. In the subsequent months, there was a concerning underreporting of cases of arboviruses, including dengue, zika, and chikungunya, both in Brazil and in other regions. This phenomenon can be explained by the prioritization of health resources to face COVID-19, as well as possible limitations in the capacity for diagnosis and notification of other infectious diseases [23,24].

Importantly, seasonality describes temporal patterns that influence the spread of diseases. Considering chikungunya, this seasonality is intrinsically linked to climatic and environmental elements, playing a predominant role in the dynamics of its transmission. Temperature variations resulting from the different seasons of the year directly impact the survival and decline of mosquito populations, the primary vectors in the transmission chain of chikungunya. This phenomenon underlines the importance of monitoring and understanding seasonal patterns, aiming at the implementation of more effective prevention and control strategies adapted to the environmental and climatic fluctuations that favor vector proliferation and, hence, virus propagation [13].

Seasonality not only directly affects the biological cycle of chikungunya vectors but also influences human behavior, altering outdoor activity patterns according to the seasons. During warmer and more pleasant periods, it is common for people to increase the time spent in outdoor environments, which raises the likelihood of contact with transmitting mosquitoes and, consequently, the risk of chikungunya transmission. This seasonal interaction between humans and vectors is a critical factor in the disease propagation dynamics, underlining the need to implement prevention strategies and awareness campaigns adapted to seasonal variations. These measures should be intensified, especially at times of the year more conducive to vector activity and human exposure behavior, with the aim of reducing the incidence of chikungunya through the education of the population about individual protection measures and vector control. This includes emphasizing the importance of using repellents, eliminating mosquito breeding sites, and installing screens on windows and doors to minimize the risk of bites [5,13,14].

Our study demonstrated a heterogeneous spatial distribution of chikungunya in Brazil, with positive spatial autocorrelation for the analyzed period from 2017 to 2023. There was a formation of high-risk spatial clusters, predominantly in the northeast region of the country. Other studies corroborate these findings, highlighting the high incidence of the disease in states and capitals of the northeast region such as Salvador [25], São Luís [26], Recife [27], Fortaleza [28], Pernambuco [29], and Maranhão [30]. This distribution pattern underlines the importance of adopting intervention and control strategies that are comprehensive and adapted to territorial specificities. Therefore, a reinforcement of human and financial resources directed towards areas identified as priorities is needed [21].

In addition, we identified that social vulnerability indicators are associated with an increase in the incidence of chikungunya in Brazil, including the social vulnerability index; individuals in households with inadequate water supply and sanitation; individuals aged 15 to 24 who do not study, do not work, and have a per capita family income equal to or lower than half a minimum wage; social vulnerability index—income and work dimension; per capita income; and the percentage of the population in households with a density >2.

Income, employment, and education are factors that influence risk areas for arbo-viruses [9]. Regardless of environmental factors, socioeconomic inequalities play a significant role in the incidence of chikungunya. A higher prevalence of the disease is observed in areas of low socioeconomic status, where disparities in access to healthcare, inadequate education, and a lack of information on preventive measures stand out as significant obstacles to disease prevention and control. Importantly, such areas are characterized by precarious housing, a lack of basic services, and limited access to healthcare, creating a scenario that favors virus transmission [18,20,21].

Demographic aspects, such as population mobility, intensify the dissemination of Chikungunya between endemic and non-endemic regions, broadening the spectrum of the epidemic. Studies indicate that the Municipal Human Development Index (MHDI) is inversely related to the occurrence of the disease, with areas of lower MHDI presenting higher incidence rates [27]. Social vulnerability, marked by income inequality and precarious living conditions, is an additional risk factor for chikungunya infection, especially in contexts where other arboviruses, such as dengue and zika, coexist, exacerbating the socioeconomic challenges faced by the affected communities [24].

Understanding the factors contributing to the emergence and spread of chikungunya epidemics in Brazil requires an integrated analysis of environmental, socioeconomic, and demographic aspects. The country’s tropical climate, characterized by high temperatures and high humidity, creates a conducive environment for the proliferation of Aedes sp. mosquitoes, the main vectors of the chikungunya virus. These climatic conditions, combined with the presence of densely populated urban areas, facilitate the rapid dissemination of the virus among the population [19].

It is crucial, however, to acknowledge the methodological limitations present in this study. The ecological nature of the research limits the ability to extrapolate results from the collective to the individual, and the use of secondary data may conceal unreported cases, potentially underestimating the true extent of the epidemic [24]. Despite these limitations, the analysis offers valuable insights into the dynamics of chikungunya, contributing to the refinement of prevention and control strategies.

## 5. Conclusions

The investigation of the chikungunya virus transmission dynamics, using spatial and temporal analysis techniques, demonstrated significant patterns related to its geographical spread in Brazil. This study highlighted that the disease’s incidence is intrinsically linked to social vulnerability indicators, taking into account the spatial aspect. The correlation found between social vulnerability and chikungunya incidence underlines the importance of such indicators as epidemiological determinants. Consequently, this methodology enabled the identification of high-risk zones, in which precarious socioeconomic conditions may amplify the proliferation of vector arthropods and hinder access to effective prevention strategies.

## Figures and Tables

**Figure 1 diseases-12-00135-f001:**
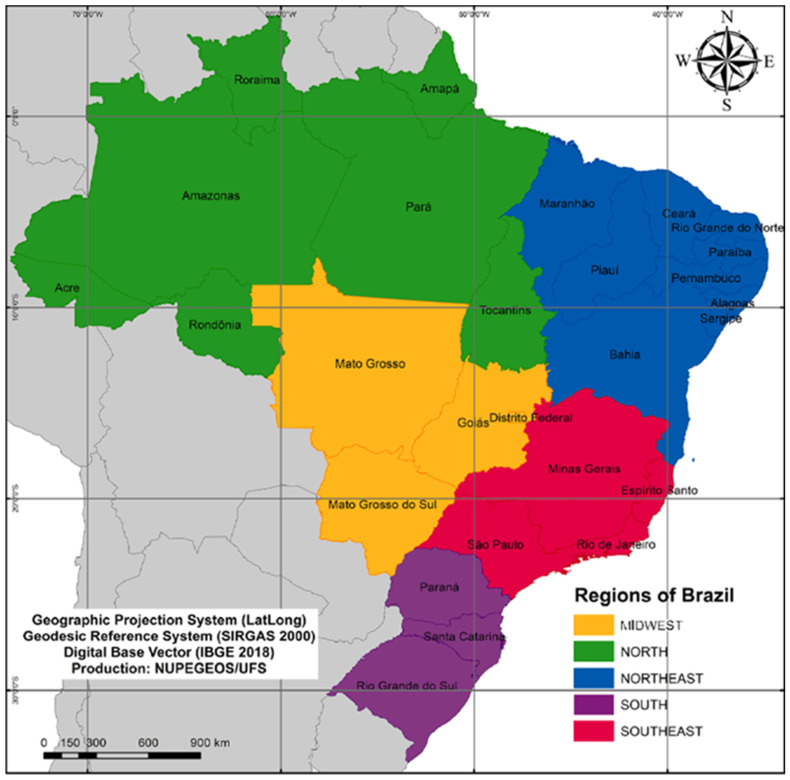
Study area (Brazil) divided into five geographic regions/states.

**Figure 2 diseases-12-00135-f002:**
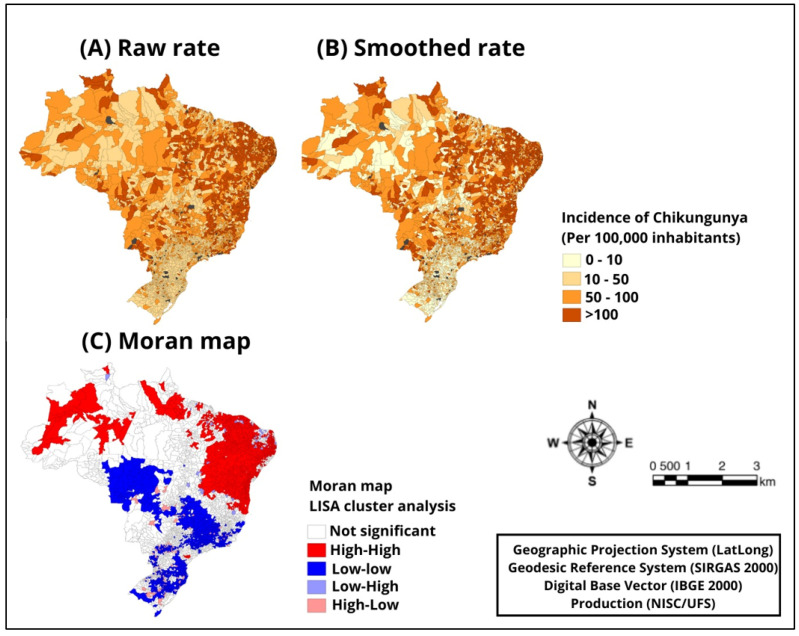
Spatial analysis of chikungunya incidence in Brazil, 2017–2023; (**A**) crude rate, (**B**) rate smoothed by the local empirical Bayesian method, and (**C**) classification of risk areas.

**Table 1 diseases-12-00135-t001:** Temporal trend analysis of chikungunya incidence, Brazil, 2017–2023.

Period	Chikungunya Incidence		
	APC (CI 95%)	*p*-Value	Trend
2017	1.90 (0.96–2.48)	<0.001	Increasing
2018–2019	−0.47 (−1.12–0.93)	0.354	Stationary
2020–2021	−0.93 (−1.18–0.98)	<0.001	Decreasing
2022–2023	−0.10 (−0.30–0.11)	0.247	Stationary

APC: Annual Percentage Change (%); 95% CI–95%: Confidence Interval.

**Table 2 diseases-12-00135-t002:** Spatial autocorrelation of chikungunya incidence in Brazil, 2017 to 2023.

Variable	Cases	Global Moran’s Index	*p*
2017	127.390	0.78	0.001
2018	50.860	0.65	0.001
2019	52.689	0.63	0.001
2020	29.339	−0.63	0.001
2021	41.817	0.67	0.001
2022	120.932	0.58	0.002
2023	64.748	0.57	0.001
2017–2023	487.775	0.80	0.001

**Table 3 diseases-12-00135-t003:** Association between chikungunya incidence and social vulnerability, Brazil, 2017–2023.

Social Vulnerability Indicators	Chikungunya	
	Rho	*p*-Value
Social vulnerability index	0.34	<0.01
Social vulnerability index—human capital dimension	0.34	<0.01
Social vulnerability index—income and work dimension	0.34	<0.01
Percentage of people in households with inadequate water supply and sanitation	0.35	<0.01
Literacy rate of the population aged 15 years and older	0.36	<0.01
Percentage of people aged 15 to 24 years who do not study, do not work, and have a per capita household income equal to or less than half the minimum wage (as of 2010).	0.28	<0.01
HDI longevity	−0.31	<0.01
HDI income	−0.33	<0.01
Per capita income	−0.33	<0.01
Vulnerable population aged 15 to 24 years	0.45	<0.01
Population in vulnerable households with elderly people	0.45	<0.01
Percentage of the population in households with a density >2	0.35	<0.01
Literacy rate—18 years or older	0.38	<0.01
Per capita income of those vulnerable to poverty	−0.33	<0.01

**Table 4 diseases-12-00135-t004:** Collinearity diagnostics between chikungunya incidence and social vulnerability indicators, Brazil 2017–2023.

Social Vulnerability Indicators	Collinearity Statistics			
	Tolerance	VIF	t	*p*
Social vulnerability index	0.066	15.175	−2.659	0.08
Social vulnerability index—human capital dimension	0.078	12.839	−2.799	0.05
Social vulnerability index—income and work dimension	0.107	9.336	4.366	<0.001
Percentage of people in households with inadequate water supply and sanitation	0.427	2.342	0.984	0.32
Literacy rate of the population aged 15 years and older	0.001	1845.373	−2.045	0.04
Percentage of people aged 15 to 24 years who do not study, do not work, and have a per capita household income equal to or less than half the minimum wage (as of 2010).	0.244	4.093	3.490	<0.001
HDI longevity	0.184	5.446	2.180	0.02
HDI income	0.035	28.749	2.899	0.04
Per capita income	0.078	12.809	−2.633	0.08
Vulnerable population aged 15 to 24 years	0.038	26.248	−3.606	<0.001
Population in vulnerable households with elderly people	0.038	26.429	4.619	<0.001
Percentage of the population in households with a density >2	0.027	3.885	0.970	0.33
Literacy rate–18 years or older	0.001	1908.591	2.576	0.01
Per capita income of those vulnerable to poverty	0.107	9.353	0.725	0.46

VIF: Variance Inflation Factor.

**Table 5 diseases-12-00135-t005:** Regression model between chikungunya incidence and social vulnerability indicators in Brazil during 2017 to 2023.

Social Vulnerability Indicators	OLS Model		Spatial Lag Model		Erro Spatial Model	
	Coefficient	*p*	Coefficient	*p*	Coefficient	*p*
Social vulnerability indicators	−2.69	<0.001	−1.28	0.003	−1.04	0.15
Social vulnerability index—human capital dimension	0.16	0.78	0.14	0.760	0.25	0.64
Social vulnerability index—income and work dimension	3.95	<0.001	1.93	<0.001	1.02	0.003
Percentage of people in households with inadequate water supply and sanitation	0.16	<0.001	0.80	<0.001	0.95	0.007
Percentage of people aged 15 to 24 years who do not study, do not work, and have a per capita household income equal to or less than half the minimum wage (as of 2010).	0.60	<0.001	0.16	0.003	−0.05	0.38
HDI longevity	−2.60	<0.001	−0.76	0.221	0.46	0.94
Per capita income	0.60	0.002	0.94	<0.001	0.14	<0.001
Percentage of the population in households with a density >2	0.79	0.003	0.58	0.004	0.12	0.006
Per capita income of those vulnerable to poverty	0.21	0.37	0.21	0.900	−0.002	0.89
**Model Evaluation Criteria**	**OLS Model**		**Spatial Lag Model**		**Erro Spatial Model**	
Determination coefficient (*p*-value)	0.58 (*p* = 0.002)		0.78 (<0.001)		0.22 (*p* = 0.008)	
Log likelihood	5.1		11.6		2.2	
Akaike criterion	49.6		26.2		78.5	
Schwarz criterion	119.1		108.5		200.6	
Moran index (*p*-value)	0.38 (*p* = 0.001)		−0.014 (*p* = 0.44)		0.12 (*p* = 0.002)	

*p*: *p*-value < 0.05.

## Data Availability

Data are contained within the article and Appendix A.

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
