# Peer review of "Spatial and Temporal Dynamics of Chikungunya Incidence in Brazil and the Impact of Social Vulnerability: A Population-Based and Ecological Study"

_diseases, 2024, doi:10.3390/diseases12070135_

Round 1
Reviewer 1 Report
Comments and Suggestions for Authors
This is an interesting paper guiding health care system to focus on high incidence foci of disease and work on variables (socioeconomic, etc) linked to increased number of cases.
The result would have been actually predicted and is compatible with other related studies in the region.
I would suggest for better data accuracy, please specify your case definition:
Confirmed cases are based on specific Chick lab test or by clinical-epidemiological criteria which is very vague. This makes me question how these cases were confirmed.
Please be more specific about definition of confirmed cases.
Comments on the Quality of English Language
Good English writing would need professional editing review.
Author Response
We appreciate the constructive comments and valuable suggestions provided by the reviewers for the article: "Spatial and temporal dynamics of Chikungunya incidence in Brazil and the impact of social vulnerability: a population-based and ecological study." These revisions have helped to enhance the quality of our manuscript.
Accordingly, we have thoroughly revised our manuscript. In response to the reviewers' comments, we have made changes in the review of writing and spelling to maintain a more fluid and clear text, corrections in text editing, as well as aspects of the methods and results.

Reviewer 2 Report
Comments and Suggestions for Authors
1. Please describe which laboratory tests were used to confirm a case.
2. Also, during the dry season, people have to storage water, becoming these bodies of water breeding sites for mosquitoes.
3. Authors area making an analysis of a eight-year period; therefore, not only seasonality but also periodicity should be included as a determinant factor in the incidence of Chikungunya. This periodicity is influenced by climatic phenomenon like: El Niño, La Niña, and the South American Monsoon System.
4. Mention possible explanations for the concentration of Chikungunya cases in the North and Northeast
5. Authors ignore that after epidemics , populations develop natural immunity that make them protected against Chikungunya.

Minor corrections mentioned in the text of the manuscript, especially italics.
Author Response

(The authors gave the same response as above.)

Reviewer 3 Report
Comments and Suggestions for Authors
In this paper, the authors study the incidence of Chikungunya in Brazil, the most affected country in the past years by evaluating its distribution among time and space. The manuscript is presented in a good way, but some questions must be addressed to increase overall paper quality:
Intro and Methods:
Introduction lacks some context for the research. It is known that Chikungunya outbreaks in Asia are characterized by years with significant increase, stability and seasonal decrease [REF]. It is known that Brazil has gaps to understand the distribution of arboviral diseases, but some papers on the subject were published in the past years. It is important to cite these papers as basis in the subject.
Why are the authors using data ranging from 2017 up to 2023? This should be clear in the text. Data from previous years is not available in Brazil?
Figure 1 shows the division by regions in Brazil highlighting its role on analysis design. However, the following analysis were done using municipalities data and it is not clear were the regions were used after that. Please update figure and sentence to provide adequate context or explicit in results text when regions are used to interpret data.
Lines 36 and 37. Species names should be italicized
Line 51. The authors must make the text more fluid and clearer, connecting related sentences. This happens many times in the text.
E.g. instead of “…especially in the Northeastern region. With the state of Ceará being the most affected.” use “…especially in the Northeastern region, with the state of Ceará being the most affected.”
Line 55. The manuscript must be reviewed for spelling errors. “Prolongated” is an incorrect word.
Line 57. “affects” instead of “affect”
Line 118. “…commonly employed in epidemiological studies.” Avoid using explanatory terms unless they are really necessary to understand the methods and results.
Lines 121 and 122. “…logarithmic function with base 10” instead of “…logarithmic function based on 10”
Line 126. The authors mention variables from an equation that is missing in the text.
Line 175. The authors do not point which studies were used to determine relevant epidemiological criteria to be analyzed.
Line 208. The APC increase was 1.90% (mentioned this way by the authors in line 278). Please correct the percentage.
Results:
The authors must show some descriptive statistics about the data used for the analysis. Either provide the dataset with supplementary material or some table resuming the number of cases found each year, their frequency among analyzed regions and other related data.
Some p-values in tables have commas as decimal separators while others have dots. Please use only dots.
The authors show the indicators correlated with CHIKV incidence but earlier in methods they mention and in text 26 total indicators to be analyzed. All the 26 indicators are not shown throughout the manuscript, only the positively correlated ones. It is important to know which indicators are not pos. correlated as this evidence could also help to shape the prospection of new policies to monitor and deal with Chikungunya outbreaks in Brazil and be used to compare with other studies.
Discussion:
The authors raise the importance of seasonality and temperatures in disease occurrence due to abundance of mosquito vectors. They discuss this major well-known association in 3 paragraphs followed by the affirmation that “Regardless environmental factors, socioeconomic inequalities play a significant role in the incidence of Chikungunya.” (line 367). Environmental and social inequalities are intimately related and this must be discussed by the authors.
Considering that states with lower incomes per capita have higher average temperatures (more suitable for vector proliferation), how the authors address the fact that different climate zones can also be related to Chikungunya incidence when studying socioeconomic factors without being confounded by climate differences among the analysis?
How most prevalent vulnerability indicators are addressed in other studies? Are they related to other aspects of disease?
The discussion lacks comparison with other studies, showing similarities among other works and how to address socioeconomic variables to future epidemiological studies in arboviruses’ outbreaks.
Comments on the Quality of English LanguageEnglish should be revised. Some words are misspelled and some senteces can be clarified.
Author Response

(The authors gave the same response as above.)
